# Gas–Water Two-Phase Flow Characteristics and Flowback Evaluation for Shale Gas Wells

**Weiyang Xie \*, Jianfa Wu, Xuefeng Yang, Cheng Chang and Jian Zhang**

Shale Gas Research Institute, PetroChina Southwest Oil & Gas Field Company, Chengdu 610051, China;
jianfa_wu@petrochina.com.cn (J.W.); yangxuefeng@petrochina.com.cn (X.Y.);
chang_cheng@petrochina.com.cn (C.C.); zhangjian420@petrochina.com.cn (J.Z.)
**\*** Correspondence: xieweiyang@petrochina.com.cn

**Abstract:** The dynamic characteristics of shale gas wells are complexly affected by the gas–water two-phase flow. Based on the special flow mechanism of gas–water two-phase flow in shale gas reservoir, this paper establishes a mathematical model for gas–water two-phase flow in shale gas multi-stage fractured horizontal wells, introduces the eigenvalue method and orthogonal transformation, and obtains the analytical solution of the two-phase flow model. The gas–water two-phase flow rules and main influence factors of shale gas wells were identified, further combined with the flowback test characteristics and data of the shale gas wells in southern Sichuan, the characteristic parameters for the evaluation of the gas well flowback effect were determined, and an index system was established for the evaluation of shale gas well flowback. The analysis result shows that the shale gas well flowback effect has a good relationship with its production capacity, which is mainly reflected in the flowback characteristic parameters such as gas breakthrough time, gas breakthrough flowback rate, 30 d flowback rate, and maximum production flowback rate. The shale gas wells with lower flowback factors have a better production capacity than those with higher flowback factors. The flowback evaluation index system can accurately forecast the shale gas well production capacity in its initial stage, and furthermore offer guidance to promptly ascertaining the block development potential and formulating the development schemes.

**Keywords:** shale gas well; gas–water two-phase flow; sensibility of two-phase flow; flowback characteristic factors; flowback evaluation

## 1. Introduction

Due to high density and low permeability, shale gas reservoirs have no natural production. In recent decades, North America has managed to obtain considerable industrial gas flow from shale gas reservoirs by horizontal well drilling technology and large hydraulic fracturing technology [1,2]. Hydraulic fracturing mainly consists of applying a strong pressure to inject the slick water (accounting for 99%) plus ceramsite, quartz sand and other proppants into the shale reservoir, and supporting the artificial fractures and natural fractures in the reservoir with the proppants as far as possible while pressing the formation open, so that the shale gas can be taken to the ground through these fracture channels [3–5]. In the earlier production period, shale gas wells mainly produce free gas in reservoir microfractures and pores. However, as the formation pressure gradually drops and approaches the critical value of the adsorption, the adsorbed gas begins to desorb. Therefore, the production in the middle and later periods is mainly from the desorbed gas [6–9]; direct gas-in-place measurements prove much higher production potential than expected for shale formations.

Since hydraulic fracturing requires the injection of a large volume of slick water (generally more than 20,000 m³ per well), generally, the production of a shale gas well in the earlier and middle periods includes water production. The concurrent flow of gas and

water has a significant impact on the production dynamic analysis and capacity evaluation of the shale gas well [10,11]. At present, studies by scholars at home and abroad on gas–water two-phase flow of the gas wells are mainly divided into two parts: experiments and calculation. In terms of experiments, in 2004, Wu Jianfa analyzed the water channeling, streaming around, and jamming phenomena of gas–water two-phase seepage flow in fractured reservoirs, specified the seepage flow mechanism of gas–water two-phase flow in fractured reservoirs, and found the seepage flow discontinuity nature of gas–water two-phase flow [12]. In 2007, Zhou Keming et al. conducted a gas–water two-phase seepage flow test study, pointing out that the deviation of the relative permeability curve obtained by calculations based on various gas–water two-phase seepage flow models is relatively large [13]. In 2012, Yan Youjun et al. conducted a study on the mechanism of driving gas with water and related laboratory experiments, and found that the formation of closed gas is mainly caused by the blocking in the driving process. If there are densely disconnected pores in the reservoir, closed gas may also be formed [14]. In terms of calculation methods, in 2010, Huang Quanhua et al. studied the low-permeability water-producing gas reservoirs and established a capacity formula for low-permeability gas-producing gas wells with consideration of the impact of starting pressure gradient and gas–water two-phase seepage flow [15]. In 2014, Yuan Lin et al. proposed a generalized gas–water two-phase pseudo-pressure function and analyzed the factors affecting the gas–water two-phase flow capacity of the low-permeability gas-producing gas wells [16]. In 2015, Liu Shudong et al. discussed the production capacity prediction methods for fishbone branch horizontal wells under gas–water two-phase flow conditions, and preliminarily determined the impact of starting pressure gradient and stress sensitivity coefficient on gas–water two-phase flow [17]. In 2016, Zhao Jinzhou et al. developed a shale gas reservoir fracturing capacity simulator based on numerical simulation and numerical calculation methods of oil and gas reservoirs, and quantitatively analyzed the impact of various geological engineering parameters on the production preliminarily [18]. In 2016, Zhang Fenna et al. established a model for the conductivity of the gas-producing channels with variable fractures and variable flows in the gas–water two-phase flow stages based on the fracturing water production of CBM wells and believed that the CBM desorption and coal matrix shrinkage will lead to significant changes in the conductivity of the gas-producing channels [19]. In 2016, Guo Xiaozhe et al. established a gas–water two-phase seepage flow model for shale gas multi-stage fractured horizontal wells by numerical simulation, and systematically analyzed the fracturing scale, desorption, Knudsen diffusion, and gas slippage effect, as well as the impact of stress sensitivity on the capacity of shale gas wells through the numerical solution of the model [20]. In 2017, Zhang Tao et al. set up a numerical simulation of the gas–water two-phase flow in shale gas wells and specified the relationship between reservoir water saturation and gas well production-level flowback rate, offering certain guidance to the development of the shale gas well production system [21].

The shale gas well undergoes a fracturing fluid flowback test stage before it is officially put into production. In this stage, the gas and liquid production capacities of the gas well can be tested by changing the size of the nozzle. The early flowback test data obtained to some extent characterize the reservoir physical properties and fracturing transformation effects, and can be used to quickly screen for effective fracturing design, determine the critical reservoir physical property, and predict long-term production. However, there are few study reports on the use of flowback test data to evaluate the capacity of shale gas wells at home and abroad.

The shale gas well dynamic analysis and productivity evaluation are challenging because of complex gas–water two-phase flow from formation to wellhead. The research method that commercial software commonly uses is a single-phase model to simulate the shale gas well dynamically, the results are significantly different to the field situation.

This paper establishes a mathematical model for gas–water two-phase flow in shale gas multi-stage fractured horizontal wells, by introducing the eigenvalue method and orthogonal transformation. An analytical solution was obtained, and the sensitivity factors

affecting the changes in production of gas–water two-phase flow were determined by numerical inversion. By the flowback phenomenon of actual shale gas wells of southern Sichuan, the relationship between various sensitivity factors and flowback index parameters was discussed, an exploratory method for gas well capacity prediction based on the initial flowback test data was proposed, and an evaluation index system for shale gas well flowback characteristic parameters that can accurately evaluate the shale gas well capacity in the early testing stages was established to obtain the production potential of the region, providing theoretical support for shale gas field development schemes.

## 2. Rules of Gas–Water Two-Phase Flow in Shale Gas Reservoirs

After a shale gas well is fractured, there will be gas–water two-phase flow in the hydraulic fracture. In the earlier period, the water production is abundant but the gas production is limited, then the gas production increases and the water production decreases. The fracturing fluid flowback rates and durations are different by the difference of shale formation conditions and fracturing processes.

The volume fracturing can effectively transform the dense and extra-low-permeability reservoir, but due to its specialty, a large volume of slick water (the proportion of slick water can reach up to 99%) will flow back from the wellhead due to pressure difference. Due to the complicated fracture structure and high tortuosity, the fractured horizontal well can only flow back 10~30% of the injected liquid. Due to the complicated reservoir conditions in some areas, the fracturing transformation of shale gas wells is imperfect, and consequently no complicated artificial fracture network is formed around the wellbore. Thus, it is difficult to lock the fracturing fluid, leading to a high flowback rate, which directly affects the production of shale gas wells. As a result, studying the concurrent flow rules of water and gas in shale reservoirs and optimizing the flowback process have gradually become a new focus in the field of shale gas study.

### 2.1. Mathematical-Physical Model for Two-Phase Flow in Fractured Horizontal Wells in Shale Gas Reservoir

By the wellbore gas–water two-phase flow characteristics, a physical gas–water two-phase seepage flow model for dual-medium multi-stage fractured horizontal wells in the shale gas reservoir was established as shown in Figure 1. To qualitatively study the rules of gas–water two-phase flows inside and outside the shale gas well, the following assumptions were made:

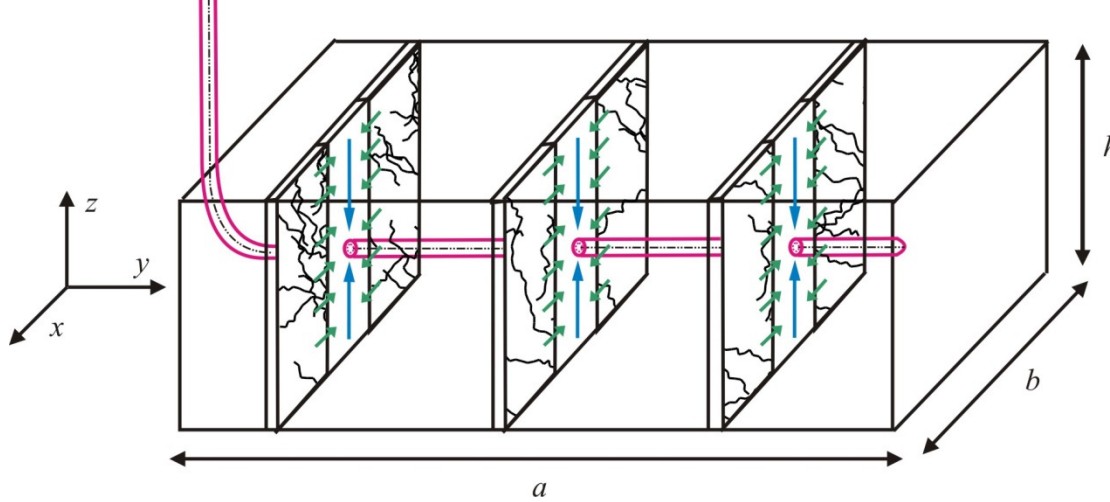

**Figure 1.** Physical model of two-phase seepage flow in shale gas fractured horizontal wells.

(1) The shale reservoir is a dual-medium reservoir, and the fluid flow in the reservoir includes the matrix desorption and diffusion and the fracture Darcy flow.

(2) The flows in the fracture system and horizontal wellbore are gas–water two-phase flows.

(3) The reservoir is penetrated by multiple non-deformable hydraulic fractures, and the hydraulic fractures are equally spaced.

(4) The flows in hydraulic fracture and horizontal wellbore are both infinite flows.

(5) The shale gas reservoir is a horizontally infinite reservoir with closed top and bottom boundaries.

According to the assumptions, a mathematical model of gas–water two-phase seepage flow in fractured horizontal wells of the shale gas reservoir was established.

(1) The matrix diffusion equation is expressed as follows:

Matrix gas-phase equation:

$$\frac{\partial \left(\rho_{\mathrm{m}} S_{\mathrm{g}} \phi_{\mathrm{m}}\right)}{\partial t} - q_{\mathrm{mm}} = 0 \tag{1}$$

Matrix water-phase equation:

$$\frac{\partial \left(\rho_{\mathrm{m}} S_{\mathrm{w}} \phi_{\mathrm{m}}\right)}{\partial t} = 0 \tag{2}$$

(2) The fracture seepage equation is expressed as follows:

Fracture gas-phase equation:

$$\nabla (\rho_{\mathrm{g}} \frac{k_{\mathrm{f}} k_{\mathrm{frg}}}{\mu_{\mathrm{g}}} \nabla p_{\mathrm{f}}) + q_{\mathrm{m}} = \frac{\partial \left(\rho_{\mathrm{g}} S_{\mathrm{g}} \phi_{\mathrm{f}}\right)}{\partial t} \tag{3}$$

Fracture water-phase equation:

$$\nabla (\rho_{\mathrm{w}} \frac{k_{\mathrm{f}} k_{\mathrm{frw}}}{\mu_{\mathrm{w}}} \nabla p_{\mathrm{f}}) + q_{\mathrm{m}} = \frac{\partial \left(\rho_{\mathrm{w}} S_{\mathrm{w}} \phi_{\mathrm{f}}\right)}{\partial t} \tag{4}$$

The hydraulic fracture two-phase flow equation is obtained:

$$\nabla (\rho_{\mathrm{g}} \frac{k_{\mathrm{frg}}}{\mu_{\mathrm{g}}} + \rho_{\mathrm{w}} \frac{k_{\mathrm{frw}}}{\mu_{\mathrm{w}}}) \nabla p_{\mathrm{f}} + \frac{q_{\mathrm{m}}}{k_{\mathrm{f}}} = \frac{\phi_{\mathrm{f}}}{k_{\mathrm{f}}} \frac{\partial (S_{\mathrm{g}} \rho_{\mathrm{g}} + S_{\mathrm{w}} \rho_{\mathrm{w}})}{\partial t} \tag{5}$$

Based on the two-phase flow theory, a gas–water two-phase pseudo-pressure function is introduced as follows:

$$m(p) = \int_{0}^{p_{\mathrm{f}}} (\frac{\rho_{\mathrm{g}} k_{\mathrm{frg}}}{\mu_{\mathrm{g}}} + \frac{\rho_{\mathrm{w}} k_{\mathrm{frw}}}{\mu_{\mathrm{w}}}) \mathrm{d} p_{\mathrm{f}} \tag{6}$$

Substituting the two-phase pseudo-pressure function into the equation above and defining the gas–water flow parameter group as $c_{\mathrm{t}} = \rho_{\mathrm{g}0} C_{\mathrm{g}} S_{\mathrm{g}} + \rho_{\mathrm{w}0} C_{\mathrm{w}} S_{\mathrm{w}}$, the pseudo-pressure form of the two-phase seepage flow equation of the fracture can be obtained:

$$\nabla^{2} m(p_{\mathrm{f}}) + \frac{q_{\mathrm{m}}}{k_{\mathrm{f}}} = \eta_{\mathrm{f}} \frac{\partial m(p_{\mathrm{f}})}{\partial t} \tag{7}$$

where

$$\eta_{\mathrm{f}} = \frac{\phi_{\mathrm{f}} c_{\mathrm{t}}}{k f \left(\frac{\rho_{\mathrm{g}} k_{\mathrm{frg}}}{\mu_{\mathrm{g}}} + \frac{\rho_{\mathrm{w}} k_{\mathrm{frw}}}{\mu_{\mathrm{w}}}\right)_{i}}$$

According to the assumptions, the seepage area of a shale gas multi-stage fractured horizontal well has closed top and bottom boundaries, and the dimensionless variables are shown in Table 1.

**Table 1.** Dimensionless parameters.

| Dimensionless Variable | Dimensionless Expression |
|---|---|
| Dimensionless two-phase pseudo pressure | $m_D = \frac{k_f h}{q_t}(m_i - m_{f/m})$ |
| Dimensionless time | $t_D = \frac{k_f t}{hL^3(\eta_f + \frac{\eta_m}{L^2})}$ |
| Two-phase flow storage ratio and channeling coefficient | $\omega = \frac{\eta_f L^2}{\sigma}, \lambda = \frac{D\rho_{sc}\eta_m h^3}{k_f}$ |
| Dimensionless fracture pseudo-pressure difference and matrix pseudo-pressure difference | $\Delta m_{Df} = m_{Di} - m_{Df}, \Delta m_{Dm} = m_{Di} - m_{Dm}$ |
| Dimensionless coordinate values in *x*, *y*, *z* directions and matrix block radius | $x_D = \frac{x}{L}, y_D = \frac{y}{L}, z_D = \frac{z}{L}, r_{mD} = \frac{r_m}{R}$ |
| Dimensionless gas equilibrium concentration and dimensionless fracture gas average concentration | $V_{aD} = V_a - V_i, V_D = V - V_i$ |
| Dimensionless gas mass density | $c_{mD} = \frac{1}{\rho_{sc}}(c_m - c_i)$ |
| Pseudo-steady state diffusion parameter group | $\sigma = \eta_m + \frac{3p_{sc}T}{T_{sc}}\frac{k_f h}{q_{sc}p_i}$ |

The dimensionless production point source in three-dimensional Cartesian coordinates is defined by the Dirac trigonometric function as follows:

$$\widetilde{q}_D = \frac{1}{q_t}(q_g B_g \rho_g + q_w \rho_w)\delta(x - x_i, y - y_i, z - z_i, t - \tau) \tag{8}$$

After Laplace transformation, Equation (7) is transformed as follows:

$$\nabla^2 \Delta \overline{m}_{Df} = \omega s \Delta \overline{m}_{Df} + (1 - \omega)\frac{h^2 L}{q_t}\overline{q}_m + \widetilde{q}_D \tag{9}$$

$$\overline{q}_{mm} = \frac{\lambda s D \rho_{sc} h}{q_t}\Delta \overline{m}_{Dm} \tag{10}$$

The expression for the pseudo-steady state diffusion at the microfracture surface is as follows:

$$q_{mf} = -G\rho_{sc}\frac{6\pi^2 D}{R^2}(V_a - V) \tag{11}$$

Introducing the Langmuir isothermal adsorption law, the expression can be obtained as:

$$\overline{V}_{aD} = -\overline{\left[\frac{q_t}{k_f h}\frac{V_m m_L}{(m_L + m_f)(m_L + m_i)}m_{Df}\right]} = -\alpha \overline{m}_{Df} \tag{12}$$

Then,

$$\overline{V}_D = \frac{u}{u + s}\overline{V}_{aD} \tag{13}$$

$$s\overline{c}_{mD} = -\frac{\alpha u s}{u + s}\overline{m}_{Df} \tag{14}$$

$$\overline{q}_{mf} = -G\rho_{sc}s\overline{c}_{mD} = G\rho_{sc}(\frac{\alpha u s}{u + s})\overline{m}_{Df} \tag{15}$$

According to the setting of the desorption and diffusion amount of hydraulic fracture system, we know:

$$\overline{q}_m = \overline{q}_{mm} + \overline{q}_{mf} \tag{16}$$

After simplifying,

$$\nabla^2 \Delta \overline{m}_{Df} = \left[\omega s + (1 - \omega)(\frac{h^2 LG\rho_{sc}}{q_t} - \frac{\lambda D^2 \rho_{sc} h}{q_t})\frac{\alpha u s}{u + s}\right]\Delta \overline{m}_{Df} + \widetilde{\overline{q}}_D \tag{17}$$

The first term on the right side of the equation above is defined as the function $f(s)$, and then the equation can be simplified as follows:

$$\nabla^2 \overline{m}_D - \overline{\overline{q}}_D = f(s)\overline{m}_D \tag{18}$$

The Laplace transformation does not solve the nonlinear problem of the gas–liquid two-phase control equation, so it is necessary to introduce the eigenvalue method and orthogonal transformation.

The expression of the eigenvalue method is as follows:

$$\begin{cases} LK = -\lambda_s \omega K, a < x < b \\ K|_{x=a} = K_x|_{x=b} = 0 \end{cases} \tag{19}$$

For the closed boundaries in all directions, a three-dimensional characteristic equation set is established as follows:

$$\begin{cases} \frac{\partial^2 E(x_D,y_D,z_D)}{\partial x_D{}^2} + \frac{\partial^2 E(x_D,y_D,z_D)}{\partial y_D{}^2} + \frac{\partial^2 E(x_D,y_D,z_D)}{\partial z_D{}^2} = -\lambda_{xyz}E(x_D,y_D,z_D) \\ \frac{\partial E(x_D,y_D,z_D)}{\partial x_D}\Big|_{x_D=0} = 0, \frac{\partial E(x_D,y_D,z_D)}{\partial x_D}\Big|_{x_D=\frac{a}{L}} = 0 \\ \frac{\partial E(x_D,y_D,z_D)}{\partial y_D}\Big|_{y_D=0} = 0, \frac{\partial E(x_D,y_D,z_D)}{\partial y_D}\Big|_{y_D=\frac{b}{L}} = 0 \\ \frac{\partial E(x_D,y_D,z_D)}{\partial z_D}\Big|_{z_D=0} = 0, \frac{\partial E(x_D,y_D,z_D)}{\partial z_D}\Big|_{z_D=\frac{h}{L}} = 0 \end{cases} \tag{20}$$

The pseudo-pressure function can be expressed as $L[m_D(x_D,y_D,z_D)] = X(x)Y(y)Z(z)$, the three-dimensional eigenvalue problem can be solved by turning it into a three one-dimensional eigenvalue problem.

In $x$ direction:

$$\begin{cases} \frac{\partial^2 E_x(x_D)}{\partial x_D{}^2} = -\lambda_x E_x(x_D), 0 < x_D < \frac{a}{L} \\ \frac{\partial^2 E_x(x_D)}{\partial x_D}\Big|_{x_D=0} = 0, \frac{\partial^2 E_x(x_D)}{\partial x_D}\Big|_{x_D=\frac{a}{L}} = 0 \end{cases} \tag{21}$$

The $x$-direction eigenvalue is $\lambda = (\beta\pi L/a)^2$ and eigenfunction is $E_x(x_D) = \cos[(\beta\pi L/a)x_D]$, $\beta = 1, 2, 3 \ldots$;

Similarly, the eigenvalues and eigenfunctions in the $y$ direction and $z$ direction can be solved.

The three-dimensional eigenvalue obtained is:

$$\lambda_{xyz} = \left(\frac{\beta\pi L}{a}\right)^2 + \left(\frac{\gamma\pi L}{b}\right)^2 + \left(\frac{n\pi L}{h}\right)^2 \tag{22}$$

The corresponding eigenfunction is as follows:

$$E_{xyz}(x_D, y_D, z_D) = \cos\left(\frac{\beta\pi L}{a}x_D\right) \cos\left(\frac{\gamma\pi L}{b}y_D\right) \cos\left(\frac{n\pi L}{h}z_D\right) \tag{23}$$

The eigenfunction constituted by Equation (23) has complete orthogonality in three-dimensional space, and the equation of complete orthogonality is:

$$F = \int_{-\infty}^{+\infty} \int_{-\infty}^{+\infty} \int_0^{\frac{h}{L}} E(x_D, y_D, z_D)E'(x_D, y_D, z_D)\mathrm{d}x_D\mathrm{d}y_D\mathrm{d}z_D \tag{24}$$

After Equation (24) is spread:

$$F = \int_0^{\frac{a}{L}} \int_0^{\frac{b}{L}} \int_0^{\frac{h}{L}} \frac{\frac{1}{6}\left[\cos\frac{\pi L}{a}x_D(\beta+\beta') + \cos\frac{\pi L}{a}x_D(\beta-\beta')\right]}{\cdot\left[\cos\frac{\pi L}{b}y_D(\gamma+\gamma') + \cos\frac{\pi L}{b}y_D(\gamma-\gamma')\right]} \; \mathrm{d}x_D\mathrm{d}y_D\mathrm{d}z_D \tag{25}$$
$$\cdot\left[\cos\frac{\pi L}{h}z_D(n+n') + \cos\frac{\pi L}{h}z_D(n-n')\right]$$

Equation (25) can be rewritten as:

$$F = \frac{1}{6} A_{\beta\beta'} B_{\beta\beta'} C_{\gamma\gamma'} D_{\gamma\gamma'} E_{nn'} F_{nn'} \tag{26}$$

The orthogonal transformation is introduced, defined as follows:

$$F[W(x)] = \int_a^b s(x) K(x, \mu) W(x) \mathrm{d}x \tag{27}$$

The two-phase seepage flow differential equation is transformed into the following form after the orthogonal transformation:

$$\left[ \left(\frac{\beta\pi L}{a}\right)^2 + \left(\frac{\gamma\pi L}{b}\right)^2 + \left(\frac{n\pi L}{h}\right)^2 \right] F[\overline{m}_{\mathrm{D}}] - \overline{\overline{q}}_{\mathrm{D}} E(x_{\mathrm{D}i}, y_{\mathrm{D}i}, z_{\mathrm{D}i}) = f(s) F[\overline{m}_{\mathrm{D}}] \tag{28}$$

According to the equation above, the pseudo-pressure expression in Laplace space based on orthogonal transformation can be obtained as follows:

$$F[\overline{m}_{\mathrm{D}}] = \frac{\overline{\overline{q}}_{\mathrm{D}} E(x_{\mathrm{D}i}, y_{\mathrm{D}i}, z_{\mathrm{D}i})}{f(s) + \left(\frac{\beta\pi L}{a}\right)^2 + \left(\frac{\gamma\pi L}{b}\right)^2 + \left(\frac{n\pi L}{h}\right)^2} \tag{29}$$

According to the orthogonal transformation definition in Equation (29), the inverse transformation formula can be obtained as follows:

$$F^{-1}[W_n] = W(x) = \sum_{n=1}^{\infty} \frac{F[W_n]}{\|E_n\|^2} E_n(x) \tag{30}$$

After a norm is introduced:

$$\|E_{nx}\|^2 = \int_a^b s(x) E_n{}^2(x) \mathrm{d}x \tag{31}$$

The conditions for the final value of the norm are: $\beta = \beta'$, $\gamma = \gamma'$, and $n = n'$, based on which the orthogonal transformation norm in this section can be expressed as follows:

$$\|E_n\|^2 = \frac{1}{6} B_{\beta\beta'} D_{\gamma\gamma'} F_{nn'} \tag{32}$$

The expression of the orthogonal inverse transformation obtained in Laplace space is as follows:

$$F^{-1}[\overline{m}_{\mathrm{D}}] = \sum_{n=0}^{\infty} \frac{F[\overline{m}_{\mathrm{D}}]}{\|E_{nx}\|^2} E_n(x) \sum_{n=0}^{\infty} \frac{F[\overline{m}_{\mathrm{D}}]}{\|E_{ny}\|^2} E_n(y) \sum_{n=0}^{\infty} \frac{F[\overline{m}_{\mathrm{D}}]}{\|E_{nz}\|^2} E_n(z) \tag{33}$$

After Equation (29) is substituted into the expression above and the expression gets simplified,

$$F^{-1}[\overline{m}_{\mathrm{D}}] = \sum_{\beta=0}^{\infty} \sum_{\gamma=0}^{\infty} \sum_{n=0}^{\infty} \left\{ \frac{6 \overline{\overline{q}}_{\mathrm{D}} E(x_{\mathrm{D}i}, y_{\mathrm{D}i}, z_{\mathrm{D}i})}{B_{\beta\beta'} D_{\gamma\gamma'} F_{nn'} \left[ f(s) + \left(\frac{\beta\pi L}{a}\right)^2 + \left(\frac{\gamma\pi L}{b}\right)^2 + \left(\frac{n\pi L}{h}\right)^2 \right]} E(x_{\mathrm{D}}, y_{\mathrm{D}}, z_{\mathrm{D}}) \right\} \tag{34}$$

$$E(x_i, y_i, z_i) = \cos\left(\frac{\beta\pi L}{a} x_{\mathrm{D}i}\right) \cos\left(\frac{\gamma\pi L}{b} y_{\mathrm{D}i}\right) \cos\left(\frac{n\pi L}{h} z_{\mathrm{D}i}\right) \tag{35}$$

$$E(x_{\mathrm{D}}, y_{\mathrm{D}}, z_{\mathrm{D}}) = \cos\left(\frac{\beta\pi L}{a} x_{\mathrm{D}}\right) \cos\left(\frac{\gamma\pi L}{b} y_{\mathrm{D}}\right) \cos\left(\frac{n\pi L}{h} z_{\mathrm{D}}\right) \tag{36}$$

where $(x_{\mathrm{D}i}, y_{\mathrm{D}i}, z_{\mathrm{D}i})$ is the fracture source point and $(x_{\mathrm{D}}, y_{\mathrm{D}}, z_{\mathrm{D}})$ is the observation point.

Equation (34) is a dimensionless pseudo-pressure expression in the Laplace space. Based on the assumption of infinite flow, the pressure anywhere in the hydraulic fracture is equal to that at any position in the horizontal wellbore, and in consideration of the impact of skin and well storage, Equation (34) can be changed as:

$$\overline{m}_{\mathrm{D}ij}(x_{\mathrm{D}i}, y_{\mathrm{D}i}, z_{\mathrm{D}i}) = \frac{s\overline{m}_{\mathrm{wD}}(x_{\mathrm{D}i}, y_{\mathrm{D}i}, z_{\mathrm{D}i}) + S_{\mathrm{K}}}{s + C_{\mathrm{D}}s^2(s\overline{m}_{\mathrm{wD}}(x_{\mathrm{D}i}, y_{\mathrm{D}i}, z_{\mathrm{D}i}) + S_{\mathrm{K}})} \tag{37}$$

According to the Duhamel principle, the dimensionless production response solution of gas–water two-phase constant pressure production of shale gas wells in the Laplace space is shown as below:

$$\overline{q}_{\mathrm{D}} = \frac{1}{s^2\overline{m}_{\mathrm{wD}}(x_{\mathrm{D}i}, y_{\mathrm{D}i}, z_{\mathrm{D}i}) + sS_{\mathrm{k}}} \tag{38}$$

With closed top and bottom boundaries, and the three-dimensional eigenvalue can be further solved as:

$$\lambda_{xyz} = \beta^2 + \gamma^2 + \left(\frac{n\pi L}{h}\right)^2 \tag{39}$$

Three-dimensional eigenfunction:

$$E_{xyz}(x_{\mathrm{D}}, y_{\mathrm{D}}, z_{\mathrm{D}}) = \mathrm{e}^{i(\beta x_{\mathrm{D}} + \gamma y_{\mathrm{D}})} \cos\left(\frac{n\pi L}{h} z_{\mathrm{D}}\right) \tag{40}$$

A complete orthogonal equation of the three directions $x$, $y$ and $z$ can be obtained based on the complete orthogonal system equations:

$$F = \int_{-\infty}^{+\infty} \int_{-\infty}^{+\infty} \int_{0}^{\frac{h}{L}} \left[2\pi\delta(\beta - \beta')\right]\left[\pi\delta(\gamma - \gamma')\right]\left[\cos\frac{\pi L}{h} z_{\mathrm{D}}(n + n') + \cos\frac{\pi L}{h} z_{\mathrm{D}}(n - n')\right] \mathrm{d}x_{\mathrm{D}}\mathrm{d}y_{\mathrm{D}}\mathrm{d}z_{\mathrm{D}} \tag{41}$$

After the orthogonal transformation is introduced and inverse transformation is performed,

$$F^{-1}[\overline{m}_{\mathrm{D}}] = \int_{-\infty}^{+\infty} \int_{-\infty}^{+\infty} \sum_{n=0}^{\infty} \left\{\frac{\overline{\overline{q}}_{\mathrm{D}}\mathrm{e}^{i\beta(x_{\mathrm{D}i}+x_{\mathrm{D}})+i\gamma(y_{\mathrm{D}i}+y_{\mathrm{D}})}}{4\pi^2 B_{nn'}\left[f(s) + \beta^2 + \gamma^2 + \left(\frac{n\pi L}{h}\right)^2\right]}\right\} \cos\left(\frac{n\pi L}{h} z_{\mathrm{D}i}\right)\cos\left(\frac{n\pi L}{h} z_{\mathrm{D}}\right)\mathrm{d}\beta\mathrm{d}\gamma \tag{42}$$

By integrating the above equation on the fracture surface $yz$, and based on the number of fractures and accumulation of the $x$-direction micro-element sections, the dimensionless pseudo-pressure expression for constant production in the final Laplace space can be obtained, and the dimensionless production expression for the constant bottom pressure production in Laplace space can be obtained by the Duhamel principle.

### 2.2. Pressure Production Characteristics and Sensitivity of Two-Phase Flow in Fractured Horizontal Wells in a Shale Gas Reservoir

Due to the research and calculation speed, most commercial software of gas well dynamic analysis usually use a single-phase model to analyze the flow characteristics from formation to wellhead, such as IHS RTA based on "single-phase gas five linear flow model" or Essca Ecrin based on "single-phase gas double porous media model". Under this situation, we established an inversion and fitting modular by programming; this modular can achieve a gas–water two-phase flow type curve drawing of shale gas well.

The dimensionless pseudo-pressure solution in the Laplace space is transformed into real space by the Stehfest numerical inversion algorithm [22], the dimensionless pseudo-pressure in real space and pseudo-pressure derivative characteristic curve of gas–water two-phase multi-stage fractured horizontal wells in the shale gas reservoir are shown in Figure 2.

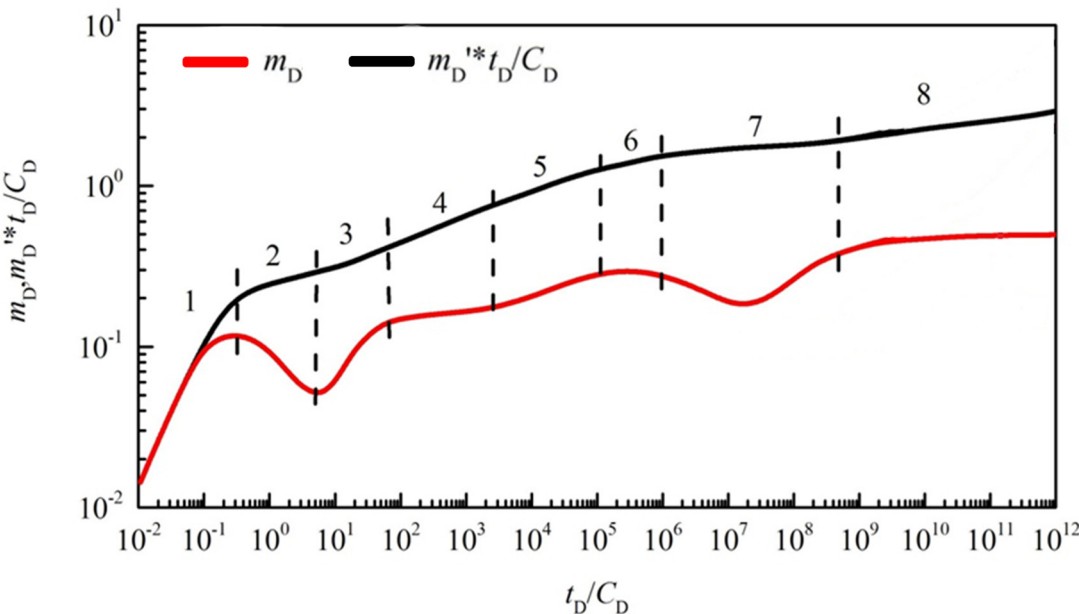

**Figure 2.** Dimensionless pseudo-pressure and pseudo-pressure derivative characteristic curve of gas-water two-phase multi-stage fractured horizontal wells in the shale gas reservoir.

Based on the Duhamel principle, the Blasingame production decline characteristic curve of the gas–water two-phase fractured horizontal wells in dual-medium shale gas reservoir with $t_{Dd}$ as the abscissa and $q_{Ddi}$ and $q'_{Ddi}$ as the vertical coordinates can be further obtained as shown in Figure 3.

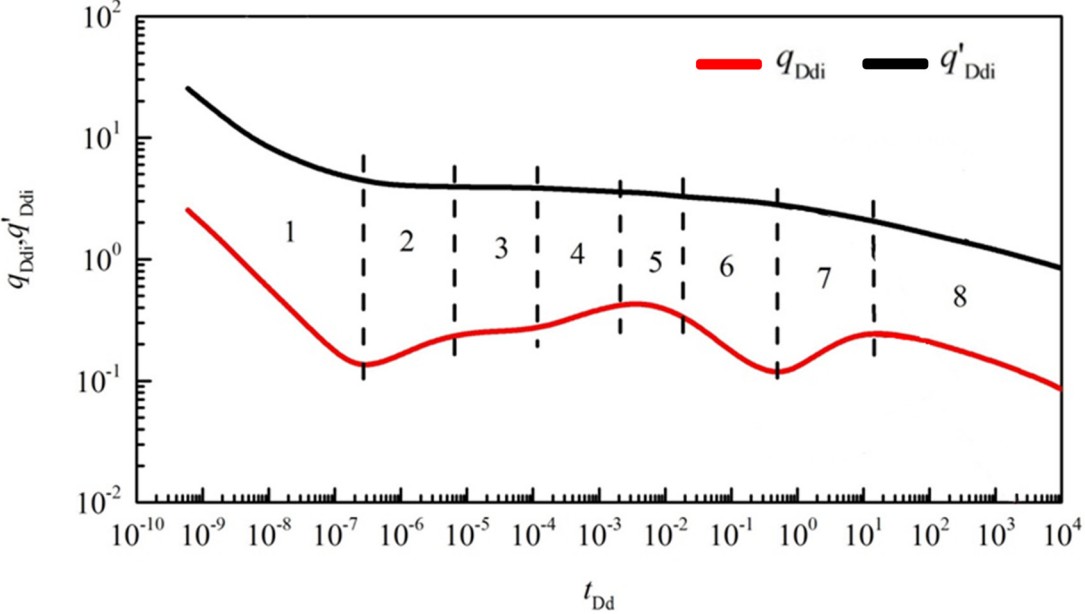

**Figure 3.** Production decline characteristic curve of gas-water two-phase multi-stage fractured horizontal wells in the shale gas reservoir.

In Figure 3, the production decline characteristic curve of the gas–water two-phase multi-stage fractured horizontal wells in shale gas reservoir can be divided into eight flow stages: ① well storage and skin impact, ② early linear flow, ③ early radial flow, ④ middle linear flow, ⑤ middle radial flow, ⑥ early matrix channeling to fracture, ⑦ late matrix channeling to fracture, and ⑧ late boundary control flow stage.

Next, the characteristic curve sensitivity of gas–water two-phase multi-stage fractured horizontal wells in the shale gas reservoir is analyzed.

### 2.2.1. Effect of Initial Gas Water Saturations $S_g$ and $S_w$

Figure 4 is the effect diagram of initial gas–water saturations on dimensionless production integrals and integral derivative curve. The increase in gas-phase saturation directly leads to the increase of the proportion of gas in the two-phase flow, and the pressure difference required to drive the gas to flow is small. The higher the gas-phase saturation, the higher the gas content, the larger the overall production, and the slower the production decline rate is, the deeper the valleys on the dimensionless production integral derivative curve are.

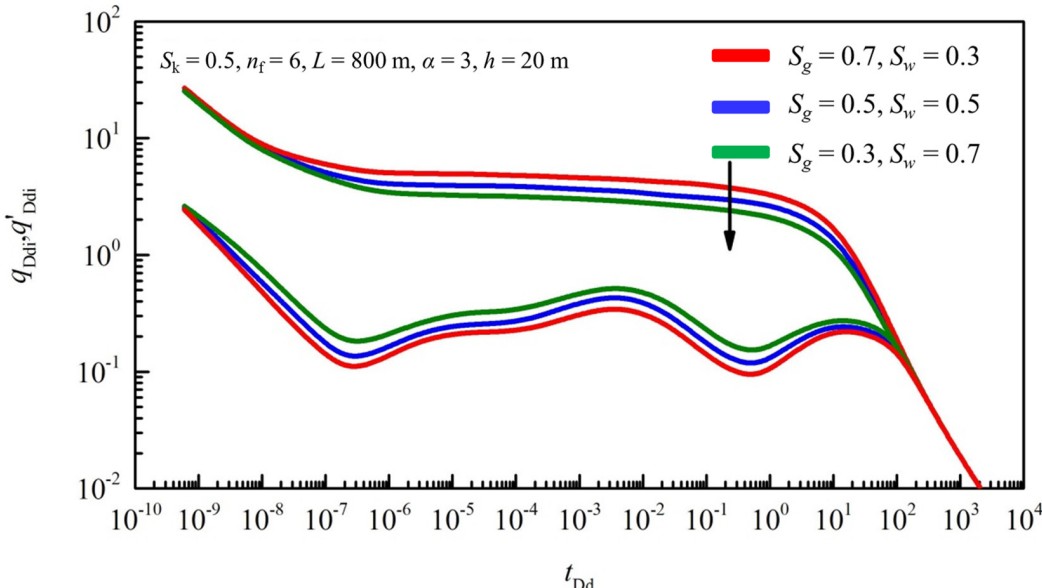

**Figure 4.** Effect of initial gas water saturations on dimensionless production integrals and derivative curve.

### 2.2.2. Effect of Storage Ratio ω

Figure 5 is the effect diagram of storage ratio on dimensionless production integrals and integral derivative curve. The effect of storage ratio on production decline is mainly divided into two stages. In the early stage of production, the fracture system has weak space storage capacity, so the storage ratio is relatively low. In the case of a gas–water two-phase flow, the small storage capacity will significantly restrain the gas and liquid co-flow capacity in the fracture, and the production decline value is relatively small. In the middle and late stages of production, the small fracture storage capacity will lead to a large production pressure difference, which will cause the desorption effect to occur in advance, resulting in a large volume of desorbed gas in microfractures. Moreover, under the action of large production pressure difference, the liquid is carried to flow, the channeling of the matrix to the fracture occurs earlier, and the production is lower. The specific performance on the curve is that the lower the storage ratio is, the earlier the dimensionless production integral derivative curve drops in the middle and late stages of flow, and the deeper valleys on the curve are.

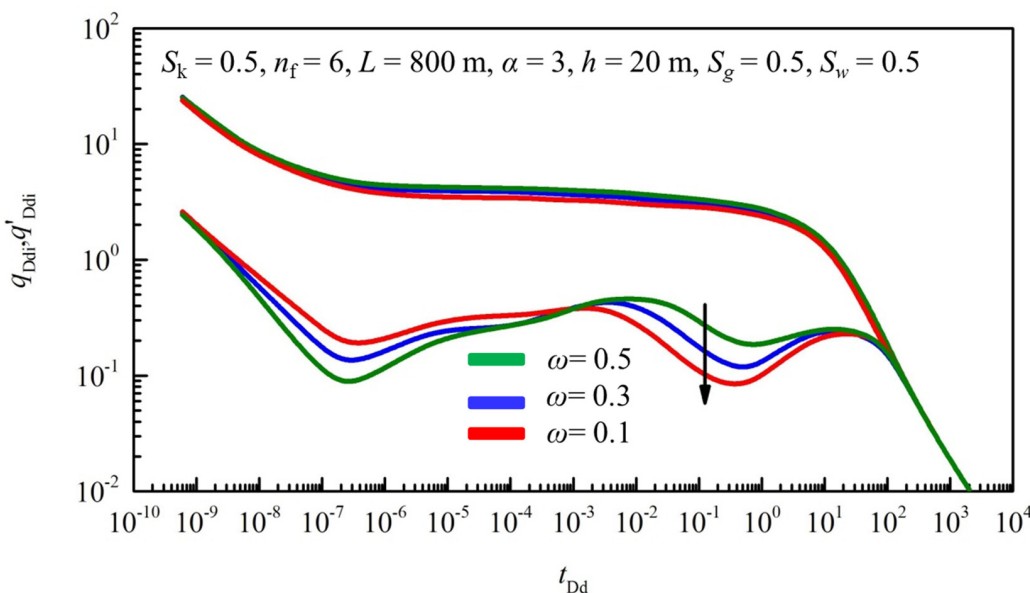

**Figure 5.** Effect of storage ratio on dimensionless production integrals and derivative curve.

### 2.2.3. Effect of Interporosity Flow Coefficient λ

Figure 6 is the effect diagram of interporosity flow coefficient on dimensionless production integrals and the integral derivative curve. The flow period mainly affected by the interporosity flow coefficient is from the middle radial flow stage to the late channeling flow stage, and the smaller the interporosity flow coefficient is, the stronger the flow capacity of the fluid in the artificial fracture is. After the free gas in the fracture is exhausted, the desorbed gas is in urgent need to supplement the deficit reservoir energy. The characteristic curve shows that the later the matrix channels to the fracture occurs, the later the dimensionless production integral derivative curve drops.

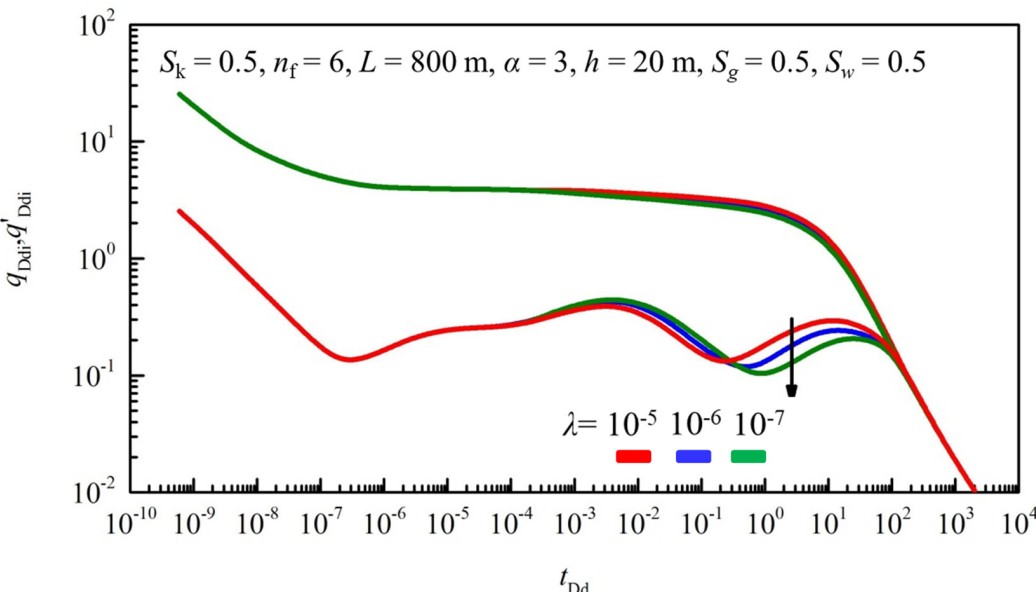

**Figure 6.** Effect of interporosity flow coefficient on dimensionless production integrals and derivative curve.

### 2.2.4. Effect of Desorption Coefficient α

Figure 7 is the effect diagram of desorption coefficient on dimensionless production integrals and integral derivative curve. A great desorption coefficient indicates that the adsorbed gas on the surface of the matrix is highly sensitive to pressure changes. The

adsorbed gas is more likely to be desorbed in the pressure difference increase process. The earlier the matrix channels, the higher the desorption coefficient is on the characteristic curve, and the higher the position of the dimensionless production integral curve is. The integral derivative curve drops before rises again.

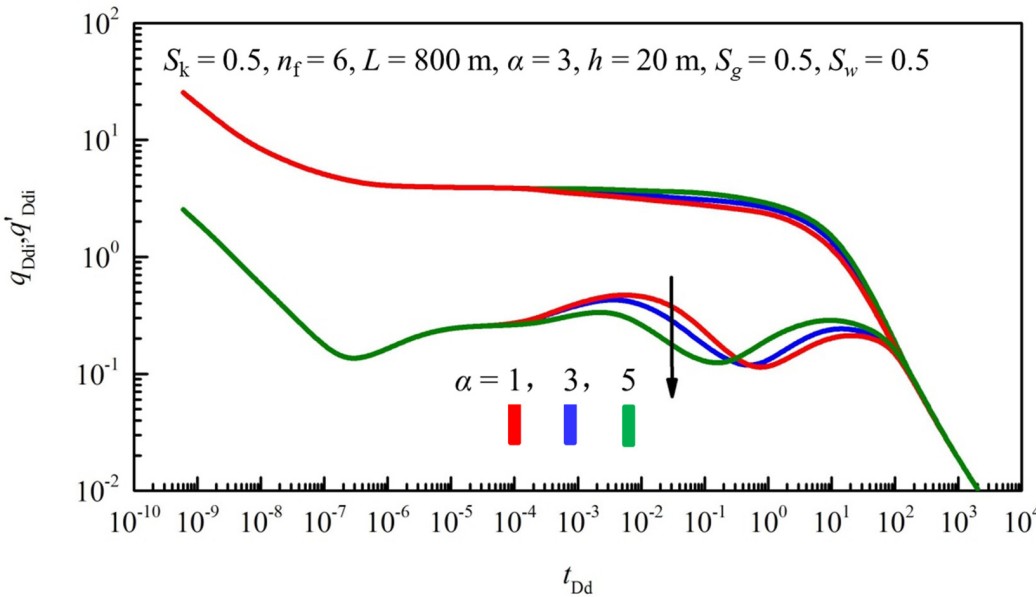

**Figure 7.** Effect of desorption coefficient on dimensionless production integrals and derivative curve.

## 3. Basic Characteristics and Characteristic Parameters of Gas–Water Two-Phase Flowback in Shale Gas Fields in Southern Sichuan

After exploration and development for nearly five years, more than 300 wells have been put into operation in the shale gas fields in southern Sichuan. Before a shale gas well is officially put into production, it undergoes a drainage period with a duration varying from one month to four months. In this period, a large volume of flowback fluid is discharged, the gas well test production can be determined, and the production data of many gas wells can be monitored and recorded to prepare for the gas well official production.

Similar to shale gas wells in North America, the shale gas wells in southern Sichuan also show the characteristics that the flowback is in a large volume in the early stage and gradually drops to close to zero in the late stage. There is a negative correlation between the volume of produced gas and the volume of flowback fluid. The more the gas is produced, the less the volume of flowback fluid is. As the production continues, the gas production gradually increases, but the volume of flowback fluid gradually decreases, and the growth rate of the flowback rate slows down. This characteristic is most evident in the initial stage of drainage. According to the characteristics of liquid production and gas production during the drainage period, the shale gas well drainage process is divided into four stages (Figure 8), whose main characteristics are as follows:

(1) The gas well begins to drain, and the liquid production gradually increases. At this stage, no gas is produced, only liquid;

(2) Weak gas flow emerges in the gas well, the liquid production increases rapidly and reaches the highest, and the gas production of the gas well gradually increases;

(3) The liquid production of the gas well decreases rapidly, and the gas production increases rapidly and reaches the highest;

(4) The liquid production of the gas well slowly decreases, while the gas production remains stable (fluctuating) and then gradually decreases. After that, the liquid production, gas production, and pressure value gradually become stable for a long time.

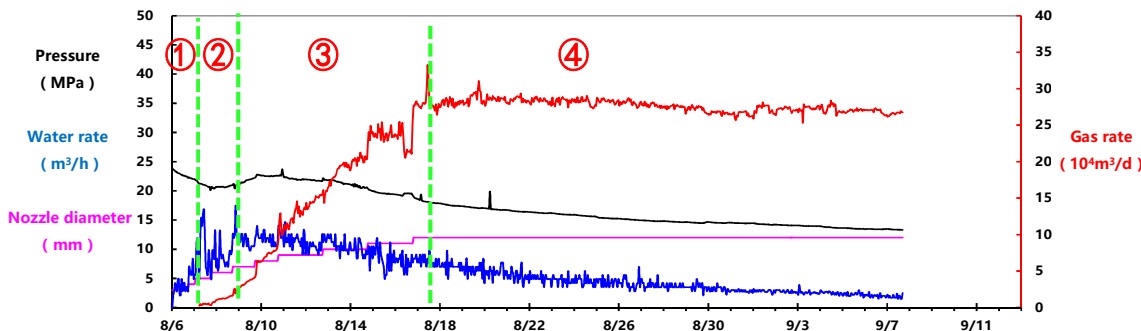

**Figure 8.** Flowback characteristic curve of shale gas wells.

According to the analysis in the previous section, the most sensitive factors affecting the production decline of shale gas–water two-phase flow are the initial gas and water saturations and storage ratio, which affect almost the entire process of shale gas well production. The changes in initial gas and water saturations are mainly reflected in the changes in production ratio of gas and liquid in the early flowback process during the shale gas well production. A high initial water saturation indicates that the volume of liquid in the large fracture and wellbore is much greater than that of the gas. This may cause high daily water production of gas wells in the early stage and occupation of flow channels by water, resulting in delayed appearance of obvious gas flow at the wellhead and higher liquid flowback rate in the early stage. The changes in the storage ratio are mainly reflected in the changes in the storage capacity of a flowing channel. If the storage ratio is high, the gas production and liquid production of the gas wells in the initial stage will quickly reach the peak, and the production decline rate is relatively high, which interacts with the initial gas and water saturations, allowing the gas production and liquid production to stabilize earlier.

According to the results of a large number of drainage tests in the North American shale gas fields, if the natural fractures in the shale reservoir are highly developed, the ground stress deviation is small, and the artificial fractures are highly extendable, the fracturing is more likely to form a complicated volume fracture system. The more complicated the fracture system is, the more obvious the water locking effect in the reservoir is, resulting in a high gas production and a low flowback rate.

The analysis on the flowback parameters and production capacities shows that the shale gas wells in southern Sichuan have similar early flowback characteristics as the shale gas wells in North America. In view of the data access difficulty and relevance, four flowback characteristic parameters were selected to evaluate the flowback effect of shale gas wells in southern Sichuan:

① Gas breakthrough time: refers to the duration from gas well opening for flowback to the emerging of an obvious gas flow at the wellhead, expressed in days;

② Gas breakthrough flowback rate: refers to the flowback rate from gas well opening for flowback to the emerging of an obvious gas flow at the wellhead, expressed in %;

③ 30 d flowback rate: refers to the flowback rate within 30 days after the gas well opening for flowback, expressed in %;

④ Flowback rate of maximum gas production: refers to the flowback rate at the time when the gas production reaches the peak after the well is opened for flowback, expressed in %.

The lower the above four characteristic parameter values are, the higher the gas well test production is, especially for the high-yield wells (Figures 9 and 10).

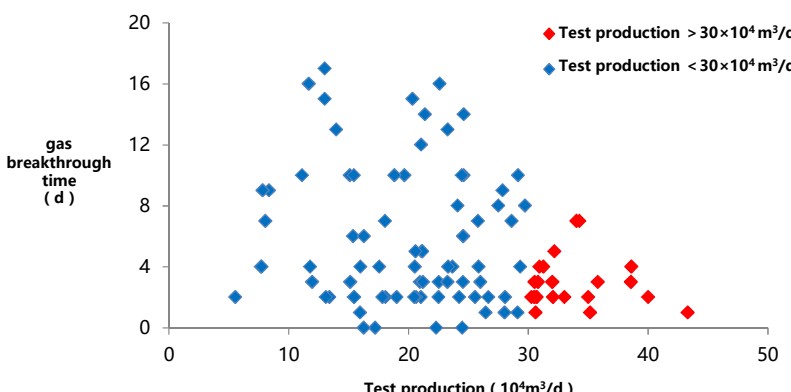

**Figure 9.** Schematic diagram of relationship between gas breakthrough time and test production of shale gas wells.

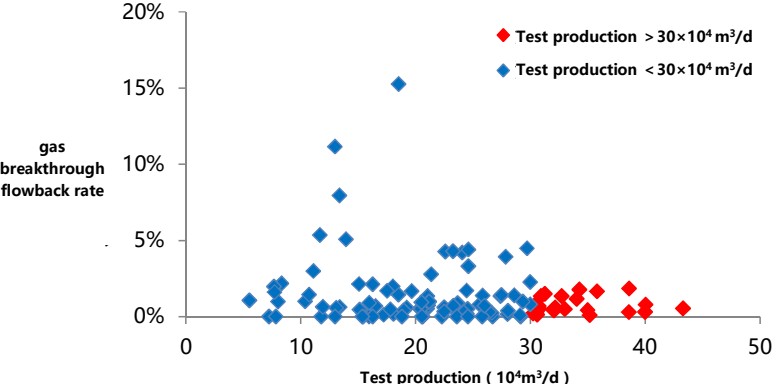

**Figure 10.** Schematic diagram of relationship between gas breakthrough flowback rate and test production of shale gas wells.

## 4. Shale Gas Well Flowback Evaluation Index System

Based on the flowback data of the production wells in Changning Shale Gasfield, a correspondence table between the shale gas well flowback parameters and production capacities was compiled as shown in Table 2 with the above four flowback characteristics parameters as the indexes.

**Table 2.** Correspondence table between flowback characteristic parameters and test production of Changning typical shale gas wells.

| Well | Test Production ($10^4$ m³/d) | Gas Breakthrough Time (d) | Gas Breakthrough Flowback Rate (%) | 30 d Flowback Rate (%) | Maximum Production Flowback Rate (%) | Production Level |
|---|---|---|---|---|---|---|
| Changning Well 1 | 27.4 | 1 | 0.23% | 8.59% | 9.08% | I |
| Changning Well 2 | 30.49 | 3 | 0.76% | 8.66% | 9.39% | I |
| Changning Well 3 | 25.56 | 1 | 0.13% | 9.24% | 9.82% | I |
| Changning Well 4 | 30.72 | 1 | 0.33% | 10.21% | 10.45% | I |
| Changning Well 5 | 25.85 | 2 | 0.19% | 9.40% | 10.64% | I |
| Changning Well 6 | 20.95 | 4 | 0.45% | 11.37% | 12.71% | I |
| Changning Well 7 | 26.72 | 2 | 0.93% | 7.56% | 8.07% | I |
| Changning Well 8 | 22.52 | 1 | 0.36% | 7.07% | 8.59% | I |
| Changning Well 9 | 20.34 | 2 | 0.54% | 7.28% | 9.68% | I |
| Changning Well 10 | 15.43 | 4 | 0.89% | 13.44% | 14.16% | II |
| Changning Well 11 | 15.47 | 3 | 0.69% | 12.03% | 12.58% | II |
| Changning Well 12 | 16.52 | 3 | 1.33% | 12.89% | 12.51% | II |
| Changning Well 13 | 13.1 | 5 | 2.21% | 7.56% | 7.73% | II |
| Changning Well 14 | 15.09 | 7 | 0.94% | 14.20% | 16.59% | II |
| Changning Well 15 | 17.22 | 7 | 1.42% | 9.40% | 9.58% | II |
| Changning Well 16 | 17.82 | 6 | 2.19% | 13.60% | 14.13% | II |
| Changning Well 17 | 7.25 | 6 | 5.23% | 17.03% | 19.42% | III |
| Changning Well 18 | 7.72 | 8 | 2.43% | 11.24% | 13.85% | III |
| Changning Well 19 | 5.55 | 11 | 3.63% | 21.66% | 26.09% | III |
| Changning Well 20 | 10.4 | 7 | 2.34% | 18.02% | 23.37% | III |

Type-I, type-II, and type-III wells in the table above are classified according to the specifications shown below.

In Table 3, type-I wells have an average gas breakthrough time of 2 days, a gas breakthrough flowback rate of 0.42%, a 30 d flowback rate of 9%, and a maximum production flowback rate of 10%, while type-II wells have an average gas breakthrough time of 8 days, a gas breakthrough flowback rate of 3.41%, a 30d flowback rate of 17%, and a maximum production flowback rate of 21%, all of which are much higher than the average values of type-I wells. There is a good correspondence between the flowback characteristic parameters and test production of typical wells.

**Table 3.** Standards for classification of gas wells in Changning shale gasfield.

| Well Type | Test Production ($10^4$ m$^3$/d) | Daily Gas Production in the First Year ($10^4$ m$^3$/d) |
|---|---|---|
| I | >20 | >10 |
| II | 10–20 | 6–10 |
| III | <10 | <6 |

A method of predicting gas well production potential with four flowback characteristic parameters was further developed. Taking the Changning Shale Gasfield as an example, according to the classification of the wells that have been put into operation (type-I, type-II, and type-III wells), the average flowback characteristic parameters of each type of well were statistically summarized, and a flowback evaluation index system for the Changning Shale Gasfield was established as shown in Table 4.

**Table 4.** Flowback evaluation indexes of gas wells in Changning Shale Gasfield.

| Well Type/Flowback Characteristic Parameters | Type-I Well | Type-II Well | Type-III Well |
|---|---|---|---|
| Gas breakthrough time (d) | ≤2 | 2~6 | >6 |
| Gas breakthrough flowback rate (%) | <1 | 1~2 | >2 |
| 30 d flowback rate (%) | <10 | 10–15 | >15 |
| Maximum production flowback rate (%) | <15 | 15–20 | >20 |

Taking the new production wells in Changning Shale Gasfield as an example, the established flowback evaluation index system was further verified (Table 5).

**Table 5.** Flowback evaluation index data of typical production wells in Changning Block.

| Well Name | Commissioning Date | Test Production ($10^4$ m$^3$/d) | Gas Breakthrough Time (d) | Gas Breakthrough Flowback Rate (%) | 30 d Flowback Rate (%) | Maximum Production Flowback Rate (%) | EUR ($10^8$ m$^3$) |
|---|---|---|---|---|---|---|---|
| Changning Well 21 | 17 January·2017 | 33.01 | 2 | 0.49% | 13.44% | 14.07% | 1.65 |
| Changning Well 22 | 29 April 2018 | 38.58 | 2 | 0.29% | 4.23% | 5.43% | 2.11 |
| Changning Well 23 | 21 July 2018 | 11.12 | 6 | 2.98% | 20.3% | 27.21% | 0.73 |
| Changning Well 24 | 10 January 2019 | 26 | 3 | 0.70% | 12.41% | 12.56% | 1.31 |
| Changning Well 25 | 13 January 2019 | 11.68 | 8 | 3.35% | 14.49% | 21.24% | 0.64 |

Changning Well 21 has a test production of 330,100 m$^3$/d, and its four flowback evaluation indexes, except for the 30 d flowback rate, reached the flowback standards of type-I wells. Its 30 d flowback rate is merely 13.44%, within the standard scope of type-I + type-II wells. Up to now, it has been in production for 2 years, with a cumulative production of more than 80 million m$^3$ and an estimated EUR of about 165 million m$^3$. Therefore, this well belongs to type-I wells and has a good production capacity and great production potential.

Changning Well 22 has a test production of 385,800 m$^3$/d, and its four flowback evaluation indexes have reached the flowback standards of type-I wells. Up to now, it has been in production for a year, with a cumulative production of more than 80 million m$^3$

and an estimated EUR of about 211 million m$^3$. Its production capacity far exceeds the average level of type-I wells and it has great production potential.

Changning Well 23 has a test production of 111,200 m$^3$/d, and its four flowback evaluation indexes have reached the flowback standards of type-III wells. It has been in production for a year, with a cumulative production of only 32 million m$^3$ and an estimated EUR of about 73 million m$^3$. Therefore, this well belongs to type-III wells with a poor production capacity.

Changning Well 24 has a test production of 260,000 m$^3$/d, and its four flowback evaluation indexes, except for the gas breakthrough time, reached the flowback standards of type-I wells. Its gas breakthrough time is merely 3 days, within the standard scope of type-I + type-II wells. Up to now, it has been in production for less than half a year, with a cumulative production of nearly 23 million m$^3$ and an estimated EUR of about 131 million m$^3$. Therefore, this well belongs to type-I wells and has a good production capacity.

Changning Well 25 has a test production of 116,800 m$^3$/d, and its four flowback evaluation indexes reached the flowback standards of type-III wells. Up to now, it has been in production for less than half a year, with a cumulative production of merely 10 million m$^3$ and an estimated EUR of about 64 million m$^3$. Therefore, this well belongs to type-III wells and has a poor production capacity.

The reliability of the current established flowback evaluation index system is verified by these examples. This is the first time the relationship between gas–water two-phase flow characteristics and well production evaluation has been established; with the flowback evaluation index system, the shale gas operator can find out the well productivity less than 3 month, even 1 month, and master the gas field development potential. After that, the decision maker can find the influence factors of shale gas well production earlier, and formulate a development adjustment plan.

## 5. Conclusions and Suggestions

### 5.1. Conclusions

This paper establishes a mathematical model based on the mechanics principle of gas–water two-phase seepage flow in shale gas wells, obtains the analytical solution of the model by eigenvalues and orthogonal transformation, further specifies the flow rules of shale gas wells in the initial gas–water two-phase flowback stage, determines four flowback evaluation characteristic parameters of shale gas wells accordingly, and establishes a flowback evaluation index system for the shale gas wells that can quickly evaluate the gas well production potential in the shale gas well flowback test stage and classify gas wells, providing a new method for promptly determining the block production potential and duly adjusting the development schemes and production rhythm. The study in the paper draws the following conclusions:

(1) The gas–water two-phase flow characteristic curve of shale gas multi-stage fractured horizontal wells can be divided into eight flow stages, and the early flowback tests of gas wells mainly affect the first five stages;

(2) Initial gas–water saturation, storage ratio, interporosity flow coefficient, and desorption coefficient are the most sensitive factors affecting the gas–water two-phase flow, of which, the first two affect all eight flow stages of the whole gas–water two-phase flow, while the latter two mainly affect the latter three flow stages;

(3) The four flowback characteristic parameters, including gas breakthrough time, gas breakthrough flowback rate, 30 d flowback rate, and maximum production flowback rate, have a good negative correlation with the gas well test production. The lower the flowback characteristic parameter values are, the better the gas well production capacity is;

(4) Based on the correspondence between the flowback characteristic parameters and test production, a flowback evaluation index system for the shale gas wells was established. The system can quickly determine the gas well production capacity in the gas well flowback test stage, providing an effective technical means for evaluating the production potential of gas wells and blocks.

*5.2. Suggestions*

The flowback evaluation index system for shale gas wells established in this paper is a useful attempt in the early flowback test evaluation of shale gas wells. Based on the results of the study and its application, the following suggestions are proposed:

(1) It is required to comprehensively collect the gas well flowback data in the initial construction and production stages of each shale gas block, evaluate the data reliability according to the difficulty and completeness of the data collection, unify the gas well production system, and select different evaluation standards for constant pressure production gas wells and production gas wells with fixed production;

(2) At least 30 sample wells are required for the establishment of the flowback evaluation index system, and it is necessary to get rid of wells with serious engineering problems (such as severe casing deformation, section loss, incomplete wellbore, etc.). The sample wells shall have as similar reservoir conditions and consistent pressure coefficients as possible;

(3) After the establishment of the flowback evaluation index system, it is necessary to continuously make real-time updates and adjustments according to the subsequent production well flowback parameters. If there are obvious changes in reservoir conditions, new process tests, etc. of the subsequent production wells, it is necessary to re-establish new flowback evaluation indexes;

(4) The mathematical model is based on gas–water two-phase flow, it can only be applied in shale gas well analysis. In shale oil wells, water is used to displace oil, the pressure boundary is quite different from shale gas well; we introduced a numerical model based on the multicomponent pseudopotential lattice Boltzmann method which can develop the oil–water flow pressure boundary conditions, and it could be very useful [23].

**Author Contributions:** Conceptualization, W.X. and J.W.; methodology, W.X.; software, W.X. and C.C.; validation, C.C. and J.Z.; resources, J.W. and X.Y.; data curation, W.X. and C.C.; writing—original draft preparation, W.X.; writing—review and editing, W.X. and J.Z.; project administration, J.W. and X.Y.; funding acquisition, J.W. and X.Y. All authors have read and agreed to the published version of the manuscript.

**Funding:** Jointly funded by "Large-scale Oil/Gas Field and Coalbed Methane Development-Changning-Weiyuan Shale Gas Development Demonstration Project" (2016ZX05062) and "Large-scale Oil/Gas Field and Coalbed Methane Development-Shale Gas Reservoir Engineering and Gas Production Process Techniques" (2017ZX05037) subordinate to the National Major Science and Technology Specific Project.

**Institutional Review Board Statement:** Not applicable.

**Informed Consent Statement:** Not applicable.

**Data Availability Statement:** Not applicable.

**Conflicts of Interest:** The authors declare no conflict of interest.

## Nomenclature

| | |
|---|---|
| $c_t$ | gas–water flow parameter group (kg/(m$^3$·MPa)) |
| $c_m$ | total mass density (kg/m$^3$) |
| $C_w$ | water-phase compression coefficient (MPa$^{-1}$) |
| $C_g$ | gas-phase compression coefficient (MPa$^{-1}$) |
| $D$ | matrix block diffusion coefficient (m$^2$/s) |
| $G$ | geometrical factor of spherical matrix block |
| $h$ | reservoir thickness (m) |
| $k$ | visual permeability (μm$^2$) |
| $k_f$ | fracture permeability (μm$^2$) |
| $k_{frg}/k_{frw}$ | gas/water relative permeability |

| | |
|---|---|
| $L$ | horizontal well length (m) |
| $m_f/m_m$ | fracture/matrix pseudo-pressure function (MPa) |
| $n_f$ | number of fractures |
| $p_L$ | Langmuir pressure (MPa) |
| $p_0$ | original formation pressure (MPa) |
| $p_{sc}$ | ground pressure (MPa) |
| $\overline{P}$ | average pressure, equal to the average value of the pressures at the inflow and outflow ends of the seepage flow unit (MPa) |
| $q_m$ | shale gas reservoir matrix diffusion flow rate (kg/(m$^3$/s)) |
| $q_t$ | gas–water two-phase characteristic mass flow rate (kg/s) at any point ($x_D$, $y_D$, $z_D$) |
| $r_m$ | radial radius (m) of spherical matrix system |
| $S_g/S_w$ | gas-phase/water-phase saturation |
| $T_{sc}$ | ground temperature (K) |
| $V_m$ | Langmuir isothermal adsorption constant (m$^3$/t) |
| $V_L$ | Langmuir volume (m$^3$/t) |
| $V_a$ | gas equilibrium concentration in pseudo-steady state diffusion (m$^3$/m$^3$) |
| $V$ | average concentration of gas in hydraulic fractures (m$^3$/m$^3$) |
| $\mu$ | fluid viscosity (mPa·s) |
| $\mu_i$ | initial gas viscosity (mPa·s) |
| $\rho_{sc}$ | natural gas density under standard conditions (kg/m$^3$) |
| $\sigma$ | desorbed gas diffusion parameter group (s$^{-1}$) |
| $\eta_f/\eta_m$ | two-phase fracture/matrix flow parameter group (s$^{-1}$) |
| $\mu_g/\mu_w$ | gas-phase/water-phase viscosity (mPa/s) |
| $\rho_g/\rho_w$ | gas-phase/water-phase density (kg/m$^3$) |

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
