# Peer review of "Gas–Water Two-Phase Flow Characteristics and Flowback Evaluation for Shale Gas Wells"

_water, doi:10.3390/w14101642_

Round 1

Reviewer 1 Report

  1. The novelty of this work is not clear,
  2. Please provide a comparison between your code with other commercial software. 
  3. Line 26: extremely is not a very appropriate word.
  4. In the first paragraph, about the gas content and desorption process: you may use: "Direct gas-in-place measurements prove much higher production potential than expected for shale formations" by Mahzari et al.
  5. A review of relative permeability and capillary pressure in shale samples would be useful. 
  6. Figure 4,5,6,7: provide proper legends indicating the curves with colour bars.
  7. Figure 8: avoid using languages other than English. Y-axis and X-axis labels are missing
  8. Figure 9 and 10: avoid using other than English

Author Response

Thank you very much for your valuable comments to our manuscript entitled “Gas-water Two-phase Flow Characteristics and Flowback Evaluation for Shale Gas Wells”. We acknowledge your comments and constructive suggestions very much.

We have made further analysis and revisions according to the your comments in detail and improved the manuscript. We hope, with these modifications and improvements based on your suggestions and comments, the quality of our manuscript can meet the publication standard of Water.

Comment 1: The novelty of this work is not clear.

Our response: The research is committed to revealing the gas-liquid two-phase flow law of shale gas wells in the early stage, and looking for the main control factors of the flowback characteristics. The established technical method can accurately evaluate the production effect of shale gas wells in the early stage, it can forecast the production potential of the block , and support the preparation of development plan. The research is rarely seen in other literature and has strong innovation and practicability.

Comment 2: Please provide a comparison between your code with other commercial software.

Our response: The analytical model to analysis of gas-water two-phase flow in shale gas wells is rare, and there is hardly any business software can realize the precise simulation of gas-water two-phase flow. The calculation code compiled in this paper can only realize the drawing, debugging, data fitting and sensitivity analysis of gas-water two-phase flow at present, other functions are still being further optimized and improved.

Comment 3: Line 26: extremely is not a very appropriate word.

Our response: We use "high-density" instead of "extremely dense".

Comment 4: In the first paragraph, about the gas content and desorption process: you may use: "Direct gas-in-place measurements prove much higher production potential than expected for shale formations" by Mahzari et al.

Our response: We add "Direct gas-in-place measurements prove much higher production potential than expected for shale formations" in the end of the first paragraph.

Comment 5: A review of relative permeability and capillary pressure in shale samples would be useful.

Our response: Due to the compactness of shale, the relative permeability curve cannot be obtained through experiments, and the capillary pressure is generally obtained through mercury injection experiments. In the paper, the relative permeability and capillary pressure are obtained through assumptions for sensitivity analysis.
Comment 6: Figure 4,5,6,7: provide proper legends indicating the curves with colour bars.

Our response: We add colour bars as legends for Figure 4,5,6,7.

Comment 7: Figure 8: avoid using languages other than English. Y-axis and X-axis labels are missing.

Our response: We have carefully checked the Figure 8, add Y-axis and X-axis labels and corrected the language mistakes.

Comment 8: Figure 9 and 10: avoid using other than English.

Our response: We have carefully checked the Figure 9 and 10, corrected the language mistakes.

We would like to take this opportunity again to express our great appreciation to the Reviewers’ comments.

Reviewer 2 Report

The authors developed a mathematical model based on mechanics principles of gas-water two-phase seepage flow in shale gas wells. They studied the effect of many factors like the storage ratio, gas water saturation and interporosity flow coefficient.

I found it difficult to follow the derivations since there are many equations, little discussion and symbols given at the end of the paper. Some equations like the Eq. 25 are hard to understand (is it a matrix or 3 terms multiplied?). The authors could include more discussion to improve the clarity.

The figures are not clear at all. Figs. 8, 9 and 10 show terms in Chinese. I suggest the authors review the figures in order to make them more clear: 1) translate terms to English; 2) Include legends with the colours (I could not understand the meaning of the different curves); 3) Clarify the different y-axes (the separation with comas for different labels is not clear); 4) Include units in those with dimensional quantities.

Do you think your mathematical model also applies to oil-water flows as in Ref. Phys. Fluids 34, 023102 (2022)?

On page 10, it is discussed the 8 stages of the production decline of the two-phase gas-water flow. Are these 8 stages general or does it depend on the sample?

Author Response

Thank you very much for your valuable comments to our manuscript entitled “Gas-water Two-phase Flow Characteristics and Flowback Evaluation for Shale Gas Wells”. We acknowledge your comments and constructive suggestions very much.

We have made further analysis and revisions according to your comments in detail and improved the manuscript. We hope, with these modifications and improvements based on your suggestions and comments, the quality of our manuscript can meet the publication standard of Water.

Comments 1: I found it difficult to follow the derivations since there are many equations, little discussion and symbols given at the end of the paper. Some equations like the Eq. 25 are hard to understand (is it a matrix or 3 terms multiplied?). The authors could include more discussion to improve the clarity.

Our response: Equation 25 is a complete orthogonality equation, which is a multiplication relationship, it indicates the rationality of orthogonal transformation of three-dimensional eigenvalue equation under boundary conditions.

Comments 2: The figures are not clear at all. Figs. 8, 9 and 10 show terms in Chinese. I suggest the authors review the figures in order to make them more clear: 1) translate terms to English; 2) Include legends with the colours (I could not understand the meaning of the different curves); 3) Clarify the different y-axes (the separation with comas for different labels is not clear); 4) Include units in those with dimensional quantities.

Our response: We have carefully checked and modified the paper, 1) Corrected the language mistakes; 2)added colour bars as legends for Figure 4,5,6,7; 3) checked the Figures, add Y-axis and X-axis labels; 4) Figure 4/5/6/7/8 have been checked, add units and Labels name.

Comments 3: Do you think your mathematical model also applies to oil-water flows as in Ref. Phys. Fluids 34, 023102 (2022)?

Our response: The mathematical model in this paper is based on gas-water two-phase flow and employ gas-water pseudo pressure to characterize the pressure dynamic characteristics of gas-water two-phase flow. The gas-water pseudo pressure includes parameters such as gas-water saturation and gas-water volume coefficient, it is not suitable for the analysis of oil-water flow.

Comments 4: On page 10, it is discussed the 8 stages of the production decline of the two-phase gas-water flow. Are these 8 stages general or does it depend on the sample?

Our response: The eight stages of the typical production decline curve of gas-water two-phase flow are suitable for idealized typical gas wells. Due to the different geological engineering conditions and production systems, stage 4 / 5 is easy to be covered up and cannot be observed in the typical production decline curve.

We would like to take this opportunity again to express our great appreciation to your comments.

Round 2

Reviewer 1 Report

Please address my previous comment "in" the revised submission, for example, about comment1, you need to discuss in the revised submission why the work is novel. About comment 2: please explain why you did not consider any commercial software. These should be discussed inside the paper. 

About Comment 4, this is a paper published in Scientific Reports, you need to properly cite the paper in the reference section. "Direct gas-in-place measurements prove much higher production potential than expected for shale formations" by Mahzari et al.

Comment 1: The novelty of this work is not clear.

Comment 2: Please provide a comparison between your code with other commercial software

Comment 4: In the first paragraph, about the gas content and desorption process: you may use: "Direct gas-in-place measurements prove much higher production potential than expected for shale formations" by Mahzari et al.

Author Response

Dear Reviewer:

Thank you very much for your letter regarding to our manuscript entitled “Gas-water Two-phase Flow Characteristics and Flowback Evaluation for Shale Gas Wells”. Thanks a lot for your reviews to our manuscript. We acknowledge your comments and constructive suggestions very much.

We have made further analysis and revisions according to the reviewers’ comments in detail and marked with Red Font in the revised manuscript. We hope, with these modifications and improvements based on your suggestions and comments, the quality of our manuscript can meet the publication standard of Water.

1.The novelty of this work is not clear.

Our response: We add detailed description and discussion about the research background and application prospect in the manuscript, marked with red font.

2.Please provide a comparison between your code with other commercial software.

Our response: In the revised manuscript we discussed the deficiency of major business software, and elaborate the use of our model inversion and fitting modular, explain the reason why we did not consider any other commercial software, and marked with red font.

3.In the first paragraph, about the gas content and desorption process: you may use: "Direct gas-in-place measurements prove much higher production potential than expected for shale formations" by Mahzari et al.

Our response: The paper “Direct gas-in-place measurements prove much higher production potential than expected for shale formations” has given us an important enlightenment that the shale gas resource has greater potential than we thought before, and many analysis methods need to be innovate and develop. We add the paper in the references.

Reviewer 2 Report

The authors have answered my questions and made the appropriate changes to the manuscript. I only have two minor comments.

The figures were certainly improved, but it is still confusing to have two y-axes and two curves or groups of curves without a clear relationship between them. For instance, in fig 2, does m_D correspond to the bottom curve and m_D'*t_D/c_D correspond to the upper curve? The same for the other figures. If yes, I suggest the authors clarify this in the caption or use different colours for the two y-axis corresponding to the colours of the curves as in fig.8.

Besides, the authors answered that this model does not apply to oil-water displacement as in Ref. Phys. Fluids 34, 023102 (2022) because the pressure assumes a different form in this case. However, they could suggest this extension of the model in the conclusion and relate to this reference. Apparently, this extension is possible and would be very useful.

After that, I believe the paper is ready for publication.

Author Response

Dear Reviewer:

Thank you very much for your letter regarding to our manuscript entitled “Gas-water Two-phase Flow Characteristics and Flowback Evaluation for Shale Gas Wells”. Thanks a lot for your reviews to our manuscript. We acknowledge your comments and constructive suggestions very much.

We have made further analysis and revisions according to the reviewers’ comments in detail and marked with Red Font in the revised manuscript. We hope, with these modifications and improvements based on your suggestions and comments, the quality of our manuscript can meet the publication standard of Water.

1.The figures were certainly improved, but it is still confusing to have two y-axes and two curves or groups of curves without a clear relationship between them. For instance, in fig 2, does m_D correspond to the bottom curve and m_D'*t_D/c_D correspond to the upper curve? The same for the other figures. If yes, I suggest the authors clarify this in the caption or use different colours for the two y-axis corresponding to the colours of the curves as in fig.8.

Our response: We carefully check all the figures, add legends in Fig.2 and Fig.3, so the two curves are more clear than before.

2. Besides, the authors answered that this model does not apply to oil-water displacement as in Ref. Phys. Fluids 34, 023102 (2022) because the pressure assumes a different form in this case. However, they could suggest this extension of the model in the conclusion and relate to this reference. Apparently, this extension is possible and would be very useful.

Our response: In “5. Conclusions and Suggestions”, we add the analysis method discuss between shale gas well and shale oil well, introduce numerical model based on the multicomponent pseudopotential lattice Boltzmann method to analysis shale oil well dynamic, and add Phys. Fluids 34, 023102 (2022) in the references.

We would like to take this opportunity again to express our great appreciation to your comments.

Round 3

Reviewer 1 Report

I have no further comment